# BCI-Walls: A robust methodology to predict if conscious EEG changes can be detected in the presence of artefacts

Bernd Porr [ORCID]☯*, Lucía Muñoz Bohollo☯

Biomedical Engineering, University of Glasgow, Glasgow, Scotland, United Kingdom

☯ These authors contributed equally to this work.
* bernd.porr@glasgow.ac.uk

## Abstract

Brain computer interfaces (BCI) depend on reliable realtime detection of conscious EEG changes for example to control a video game. However, scalp recordings are contaminated with non-stationary noise, such as facial muscle activity and eye movements. This interferes with the detection process making it potentially unreliable or even impossible. We have developed a new methodology which provides a hard and measurable criterion if conscious EEG changes can be detected in the presence of non-stationary noise by requiring the signal-to-noise ratio of a scalp recording to be greater than the SNR-wall which in turn is based on the highest and lowest noise variances of the recording. As an instructional example, we have recorded signals from the central electrode Cz during eight different activities causing non-stationary noise such as playing a video game or reading out loud. The results show that facial muscle activity and eye-movements have a strong impact on the detectability of EEG and that minimising both eye-movement artefacts and muscle noise is essential to be able to detect conscious EEG changes.

## Introduction

Brain computer interfaces (BCI) have broadly the task to turn brain activity (EEG) into actions, for example, to control a character in a video game or a wheelchair. In the simplest way this is achieved by testing if the EEG signal has reached a certain threshold [1]. However, it is well known that noise originating from muscle activity, eye-movements and other movement artefacts can interfere with the detection of conscious EEG changes [2]. Various techniques have been devised to minimise the effect of noise, in particular the independent component analysis (ICA) for offline processing [3] but for realtime closed-loop applications such as BCI one needs to resort to direct causal filtering techniques using bandpass filters, the short-time Fourier Transform, wavelet transform [4–6] or the derivative [7]. All these approaches are inferior to the offline noise removal techniques such as ICA and even after filtering the remaining noise will substantially interfere with the detection process.

Given the limited effectiveness of noise reduction techniques for BCI applications, electrode measurements will be substantially contaminated with noise, in particular EMG, because of the overlap in EEG and EMG frequencies, no matter what kind of pre-filtering has been

study is available from Zenodo: https://doi.org/10.5281/zenodo.7852162.

**Funding:** This work was supported in part by the School of Engineering, University of Glasgow.

**Competing interests:** B.P. is CEO of Glasgow Neuro LTD which manufactures the Attys DAQ board. This does not alter our adherence to PLOS ONE policies on sharing data and materials.

applied. Consequently, the detection process itself needs to cope with the noise and minimise its impact. The standard solution to maximise the signal and minimise the noise during detection is *averaging* which can be achieved in both the time domain and frequency domain:

- Time-domain: In the "evoked potential" (ep) paradigm the EEG is stimulus-locked and assumes that EMG noise is uncorrelated to the stimulus repetition and averages out.

$$\text{ep}[m] = \frac{1}{N}\sum_{n=0}^{N} d[m + n \cdot N] \tag{1}$$

where $N$ is the number of stimulus repetitions and their responses registered as $d[m + n \cdot N]$. The more repeated stimuli $N$ are presented the more the EMG noise is reduced. For example in a P300 speller a subject looks at a flashing "A" and the EEG is then added over and over again until a threshold has been reached. The more repetitions of the letter "A" the better the signal-to-noise ratio (SNR) but the longer the time to bring it over a threshold to decide the user has looked at the flashing "A".

- Frequency-domain: Here, the idea is that the subject can consciously reduce (or sometimes increase) the power of a narrow frequency band. To detect this change the signal is analysed in the frequency domain. If the band-power of a frequency band reaches a certain threshold then an action can be triggered for example moving a cursor. Again, averaging takes place because the Fourier Transform or a bank of bandpass filters accumulate the correlation between sine/cosine-waves ($e^{-j2\pi kn/N}$) and chunks of EEG $d[n]$:

$$X[k] = \sum_{n=0}^{N-1} d[n] \cdot e^{-j2\pi kn/N} \qquad k = 0, 1, 2, \ldots, N-1 \tag{2}$$

where $N$ is the number of samples the averaging takes place, $d[n]$ is the EEG and $X[k]$ its spectrum. If the chunk of EEG is long then the frequency spectrum will deliver clear peaks in the band of interest and thresholding becomes more and more reliable.

No matter if the detection process is performed in the time- or frequency-domain one needs to wait for $N$ samples until a decision can be made so that a signal reaches a threshold. Fig 1A shows such a case where a cartoon signal is shown where a signal has to reach the threshold $\gamma$ which then can be used to control for example a cursor in a BCI game. The threshold $\gamma_0$ is chosen in a way that the noise with noise variance $\sigma_0^2$ does not reach the threshold but the desired conscious EEG signal does (indicated with the tick symbol).

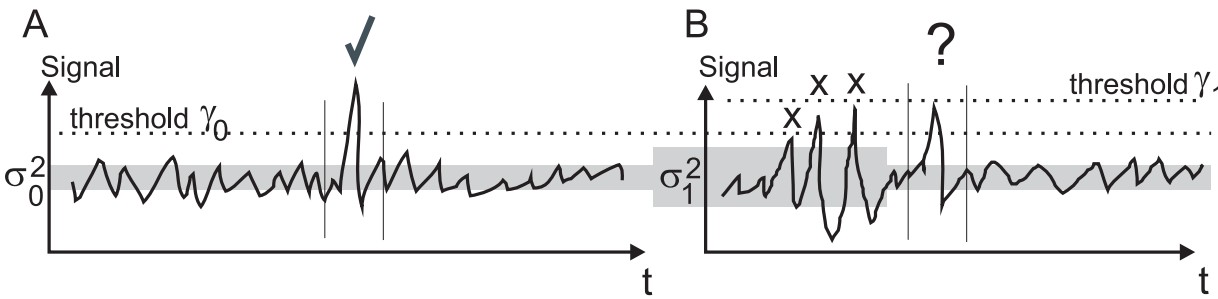

**Fig 1. Effects of stationary versus non-stationary noise on signal detection.** A) Signal detection with stationary noise, B) signal detection with non-stationary noise.

In classical detection theory there is always a number of samples *N*, if averaged over these, where reliable detection is possible. However, this does not take into account *non-stationary* noise which is the case when recording EEG contaminated with EMG, eye-movements and movement artefacts (Fig 1B). Here, the noise variance changes, for example, from the small noise variance $\sigma_0^2$ to a large noise variance $\sigma_1^2$ and then back to the small noise variance $\sigma_0^2$. The three peaks indicated at "X" are noise peaks. The threshold $\gamma_1$ could be set higher than $\sigma_0^2$ so that the 2nd and 3rd noise peaks are not detected. However, now we are encountering a problem at "?" which could be either noise or signal. It is most likely an actual signal as the noise variance has just dropped again to a lower level but this can only be known because noise levels have been asserted *a priori*. One could argue again that averaging over more samples *N* will eventually average out the noise but because of the changing noise variance peaks as the "?" become *ambiguous* and can either be signal or noise. Analytically this means that with non-stationary noise the number of samples *N* required to detect robustly a signal can reach *infinity* ($N \rightarrow \inf$) and thus detection is not possible at all. This condition is called *SNR-wall* [8].

Facial EMG and eye-movements are certainly non-stationary noise sources—in particular in everyday situations when subjects walk, play a video game or simply talk. This means that any activity a subject does will define a hard SNR-wall because the EMG and eye-movements are non-stationary. If the SNR is below the SNR-wall it is not possible to detect the EEG at all and thus it is not possible at all to design a BCI system for that kind of activity.

In this paper we introduce the concept of *SNR-walls* to EEG measurements to have an objective way to determine if an experiment can detect EEG in principle in the presence of non-stationary noise. As an instructional example of how to generally calculate SNR-walls, we have recorded the EEG from subjects during a range of different tasks. We will then calculate the SNR-walls for the different activities and conduct a statistical analysis to gauge if it's possible to construct a BCI system given the non-stationary noise created by these tasks.

## Methods

### SNR-walls

In this section, we briefly explain the relevant analytics of SNR-walls and how to calculate them practically. Tandra et al [8] provide the analytical derivation of the methodology and while they applied it to the telecommunication domain we apply it to BCI.

Let us consider a signal measured with an electrode $\tilde{d}[n]$ placed on the head of a subject:

$$\tilde{d}[n] = \underbrace{\tilde{a}[n] + \tilde{b}[n]}_{\tilde{r}[n]} + \tilde{c}[n] \tag{3}$$

where $\tilde{c}[n]$ is the consciously controlled part of the EEG. $\tilde{b}[n]$ is the background EEG activity which the subject cannot control. $\tilde{a}[n]$ are all artefacts such as muscle activity added to the measurement. Important for this paper is that the noise sources $\tilde{a}[n]$ and $\tilde{b}[n]$ are both non-stationary: for example when smiling the power of $\tilde{a}[n]$ will be larger and when relaxing the power of $\tilde{a}[n]$ will be less and non-stationary muscle noise is even generated during pure mental tasks [9]. The only stationary noise in these recordings comes from the measurement equipment. Together $\tilde{b}[n]$ and $\tilde{a}[n]$ form the noise $\tilde{r}[n]$ of the signal measured. Generally, the raw EEG signal from the electrode requires filtering, which includes at least DC and powerline interference removal but usually also bandpass filtering if only a certain EEG band is of interest:

$$d[n] = (\tilde{r}[n] + \tilde{c}[n]) * f[n] \tag{4}$$

where we subsume all filtering in $f[n]$ and from now on signals without the $\tilde{}$ are the ones after filtering.

The decision problem can be considered as a binary hypothesis testing problem which can be written as:

$$H_0 : d[n] \quad = \quad r[n] + 0 \tag{5}$$

$$H_1 : d[n] \quad = \quad r[n] + c[n] \tag{6}$$

The hypothesis $H_0$ represents the situation where the signal at the electrode contains just noise and $H_1$ where consciously generated EEG is present.

Central to the BCI system is that it is able to detect the conscious EEG $c[n]$ so that an action can be generated, for example, steering a wheelchair. Thus, we need a detector which takes the signal $d[n]$ and decides with a threshold $\gamma$ if it has just been noise or consciously generated EEG on top of the noise.

If one knows nothing about the signal except that it will increase or decrease one can just detect the average power over $N$ samples and gain a test statistic:

$$T(X) = \frac{1}{N}\sum_{n=1}^{N} x[n]^2 \tag{7}$$

which creates a random variable $T$ which in turn is then compared against a threshold $\gamma$ which decides if the signal has contained the consciously controlled EEG component $c[n]$ or not.

Since the SNR-wall analytics is quite complex we follow the same gradual approach Tandra et al used in their paper [8] and present the analytics in two steps: first, we assume that the noise is stationary which leads to standard detection theory and then in the 2nd step, we extend these equations by taking into account non-stationary noise which leads to our new BCI-walls methodology.

Let us first assume that our total noise/artefacts $r[n] = a[n] + b[n]$ have a single nominal variance $\sigma_r^2$ which means that we have stationary noise. We calculate our detection probabilities $P(D)$ as:

$$P(D)|H_0 \quad \sim \quad \mathcal{N}\left(0 + \sigma_r^2, \frac{1}{N}2(0 + \sigma_r^2)^2\right) \tag{8}$$

$$P(D)|H_1 \quad \sim \quad \mathcal{N}\left(T(c) + \sigma_r^2, \frac{1}{N}2(T(c) + \sigma_r^2)^2\right) \tag{9}$$

where $c$ is the consciously controlled EEG power.

The SNR of the signal $d(n)$ can be expressed as:

$$SNR = \frac{T(c)}{\sigma_r^2} \tag{10}$$

where $T(c)$ is the average EEG power of the consciously generated EEG component $c[n]$ and $\sigma_r^2$ is our nominal noise variance (i.e. noise power).

The probability of a false alarm $P_{FA}$ can be written as:

$$P_{FA} = \mathcal{Q}\left(\frac{\gamma - \sigma_r^2}{\sqrt{\frac{2}{N}\sigma_r^2}}\right) \tag{11}$$

where $N$ is the number of samples, $\sigma_r^2$ the noise power, and $\gamma$ the detection threshold.

In a similar way, the probability for detection $P_D$ is given by:

$$P_D = \mathcal{Q}\left(\frac{\gamma - (T(c) + \sigma_r^2)}{\sqrt{\frac{2}{N}(T(c) + \sigma_r^2)}}\right) \tag{12}$$

By eliminating the detection threshold $\gamma$ in Eqs 11 and 12 and with the help of Eqs 7 and 10 we are able to obtain an equation for the number of samples $N$ required to detect robustly the conscious EEG component $c$:

$$N = \frac{2[\mathcal{Q}^{-1}(P_{FA}) - \mathcal{Q}^{-1}(P_D)(1 + SNR)]^2}{SNR^2} \tag{13}$$

which means that at a constant noise power $\sigma_r$ one just needs to average over $N$ timesteps (Eq 7) to obtain a robust detection and with sufficiently large $N$ it is always possible to perform a robust detection.

However, brain recordings are contaminated by non-stationary noise and we need to extend the above analytics to take into account the random changes of the noise variance. To capture this noise uncertainty we define the parameter $\rho$ which is the ratio between the largest noise variance $\sigma_{r,\max}^2$ and the smallest one $\sigma_{r,\min}^2$:

$$\rho = \sqrt{\frac{\sigma_{r,\max}}{\sigma_{r,\min}}} \tag{14}$$

Our nominal noise variance $\sigma_r^2$ is then defined as:

$$\sigma_{r,\min}^2 = \frac{1}{\rho}\sigma_r^2 \tag{15}$$

$$\sigma_{r,\max}^2 = \rho\sigma_r^2 \tag{16}$$

Having defined the noise uncertainty we can express our false alarm and detection probabilities as a function of $\rho$:

$$P_{FA} = \mathcal{Q}\left(\frac{\gamma - \rho\sigma_r^2}{\sqrt{\frac{2}{N}\rho\sigma_r^2}}\right) \tag{17}$$

$$P_D = \mathcal{Q}\left(\frac{\gamma - \left(T(c) + \frac{1}{\rho}\sigma_r^2\right)}{\sqrt{\frac{2}{N}}\left(T(c) + \frac{1}{\rho}\sigma_r^2\right)}\right) \tag{18}$$

In a low SNR environment, $SNR + 1 \approx 1$ can be assumed and Eqs 7, 17 and 18 combined yield [8]:

$$N = \frac{2[\mathcal{Q}^{-1}(P_{FA}) - \mathcal{Q}^{-1}(P_D)]^2}{\left[SNR - \left(\rho - \frac{1}{\rho}\right)\right]^2} \tag{19}$$

where $N$ is again the number of samples that are needed to achieve the target probability of a permitted false alarm and probability of detection. However, now we note that when the SNR decreases the required integration/averaging steps $N$ become infinite at $SNR = (\rho - \frac{1}{\rho})$. This critical SNR is called the *SNR-wall*. Since only the denominator of Eq 19 counts, the SNR wall can be calculated simply as:

$$SNR_{wall} = \rho - \frac{1}{\rho} \tag{20}$$

The SNR-wall represents a fundamental limitation of detection ability, which means that relevant uncertainties cannot be countered by a longer averaging time. In order to determine if EEG can be detected or not we need to determine both the noise uncertainty $\rho$ and the SNR of our measured brain signal.

Having now derived the underlying analytics we can devise a step-by-step guide on how to determine if EEG changes $c[n]$ can be detected at all:

1. **Calculate the SNR-wall** by using the minimum noise variance $\sigma_{r,\min}^2$ and the maximum noise variance $\sigma_{r,\max}^2$ of the brain recording (Eq 20).

2. **Calculate the SNR** as the ratio of pure consciously controlled EEG power and noise power (Eq 10).

3. **Compare the SNR with the SNR-wall**:

$$\text{conscious EEG changes are detectable} = \begin{cases} \text{yes} & SNR > SNR_{wall} \\ \text{no} & \text{otherwise} \end{cases} \tag{21}$$

In the following sections we call the entire three-step process of determining if conscious EEG can be detected "BCI-Wall".

Having described the analytical derivations of the BCI-wall in detail we now show how to apply it in practise. This also acts as an instructional example of how to determine the BCI walls for other experiments.

## Data acquisition

As shown above, in order to determine if conscious EEG changes can be detected one needs to calculate the SNR-wall and the SNR. The SNR-wall depends on the *ratio* between minimum

noise variance and maximum noise variance (Eq 20) where different activities will cause different noise variances. For example lying down with eyes closed will have quite similar maximum and minimum noise variances resulting in a low SNR-wall. However, while reading out loud facial muscles will create short bursts of strong muscle noise as well as eye movement artefacts and the ratio between smallest and highest noise variance will be large which in turn will result in a high SNR-wall. Consequently, we have devised different tasks ranging from low noise variance ratios while lying down to extreme noise variance ratios during jaw clench. The resulting SNR-wall is then compared with the SNR. This is calculated by using the P300 evoked potential as a measure of the consciously generated EEG power and dividing it by the nominal noise power. We are now describing the experimental procedure.

Data was obtained from 20 healthy participants (9 males, 11 females). Prior to the experiment, participants were given an information sheet and were asked to give signed consent by signing two consent forms, one for the researchers and another for them to keep. Ethical approval was given by the ethics committee at the Institute of Neuroscience and Psychology, School of Psychology at the University of Glasgow, with reference 300210055. The data was acquired using an Attys data acquisition device (www.attys.tech), made up of 2 channels at 24-bit resolution, and its data acquisition programmes 'attys-ep' and 'attys-scope'. EEG recordings were single channel between an Ag/AgCl electrode at Cz & A2 and GND at A1. The 2nd channel was not used.

Participants 2 and 6 had to be excluded from the study because of faulty electrodes. About 180 sets of data are analysed in this study. These are stored in an open-access database [10], all ethics files can also be found on the database.

Data was recorded during 10 activities and P300 where pink horizontal stripes were produced randomly after 7 to 13 seconds while otherwise a chequerboard was inverted every second. The P300 was recorded for a duration of 5 minutes while the participants sat on a chair directly opposite a screen. The tasks generating various levels of artefacts were jaw clenching, reading, colouring, attempting a word search, trying a Sudoku, playing 'Subway Surfers' game on the phone, lying with eyes closed and repeat with eyes opened. Subjects sat for all tasks except the lying down ones. Each of these ran for a duration of 2 minutes. The 'attys-ep' programme was used to record the P300, while the rest were recorded using 'attys-scope'.

## EEG pre-processing

The raw EEG data $\tilde{d}[n]$ undergoes causal filtering prior to detection. Here, we have subsumed all pre-processing steps in the filter function $f[n]$. We are presenting five different filtering setups:

1. Wideband detector with minimal filtering: $f[n]$ = 0.1 Hz 4th order highpass and 50 Hz powerline bandstop which preserves the entire EMG spectrum and also low frequency fluctuations such as eye-movements. This pre-processing is hardly used in practise but acts here as a worst-case scenario as the whole EMG spectrum is allowed to interfere with the detection.

2. Low frequency bandpass filtering between a 0.1–3 Hz 4th order Butterworth filter and 50 Hz powerline bandstop. This frequency range is in the EEG delta frequency range and is in particular suitable to investigate the effect of low frequency artefacts on the ability to detect EEG such as eye-blink, eye-movements and movement artefacts from the cables.

3. Wideband bandpass: $f[n]$ = 4th order Butterworth bandpass 8–18 Hz, DC-removal, 50 Hz powerline bandstop with a moderate rejection of higher EMG frequencies used for motor imagination [11].

4. Narrow bandpass: $f[n]$ = 4th order Butterworth bandpass 8–12 Hz, DC-removal, 50 Hz powerline bandstop detecting alpha power around 10 Hz and rejects the higher frequency EMG power [12].

5. Difference operator: $f[n] = f[n] - f[n-1]$ and 50 Hz powerline bandstop which has been used for example in [7].

These five post-processing scenarios are applied separately to the data of all subjects and tasks, except for those with obvious broken electrode signals or strong artefacts.

## BCI-wall calculation

As outlined above, the BCI-wall calculations require three steps: 1) SNR-wall calculation, 2) SNR calculation and 3) comparing SNR and SNR-wall to determine if conscious EEG detection is possible. We are now describing how this can be done practically and is also an instructional example for other datasets. The corresponding Python code is available on GitHub [13].

**Step 1: SNR-wall.** In order to calculate the SNR-wall we need to calculate $\rho$ (Eq 14) which is the *ratio* between the maximum $\sigma_{r,\max}^2$ and minimum $\sigma_{r,\min}^2$ noise power. To find these two values we use a sliding window of length $\tau = 2$ sec and calculate the noise power for different sample positions:

$$\sigma_{\text{chunk}}[n] = T(d[n \ldots n + \tau]) \tag{22}$$

and obtain its maximum and minimum over the entire recording of $N$ samples: (see Fig 2):

$$\sigma_{r,\min}^2 = \min_{n=0\ldots N-1} \sigma_{\text{chunk}}[n] \tag{23}$$

$$\sigma_{r,\max}^2 = \max_{n=0\ldots N-1} \sigma_{\text{chunk}}[n] \tag{24}$$

With Eqs 14 and 20 one can then calculate the SNR-wall which is usually expressed in dB (to make it comparable to the SNR which is calculated in the next section):

$$SNR_{\text{wall}} = 10 \log_{10}\left(\rho - \frac{1}{\rho}\right) \tag{25}$$

**Step 2: SNR.** The SNR is calculated as a ratio between signal power $T(c)$ and the nominal noise power $\sigma_r^2$:

$$SNR = 10 \log_{10}\left(\frac{T(c)}{\sigma_r^2}\right) = 10 \log_{10}\left(\frac{T(c)}{\rho \sigma_{r,\min}^2}\right) \tag{26}$$

where we can re-use the SNR-wall parameters $\rho$ and $\sigma_{r,\min}^2$ from the previous section to calculate the nominal noise power $\sigma_r^2$. Measuring the pure conscious changes of EEG $c$ and its power $T(c)$ is only indirectly possible as it is not ethical to paralyse subjects. The P300 evoked potential offers an individual estimate of the consciously generated signal power $T(c)$ calculated from the peak P300 voltage. This will be used for estimating the power of the signal for both a time domain task (i.e. P300) and frequency domain task (i.e. motor imagination):

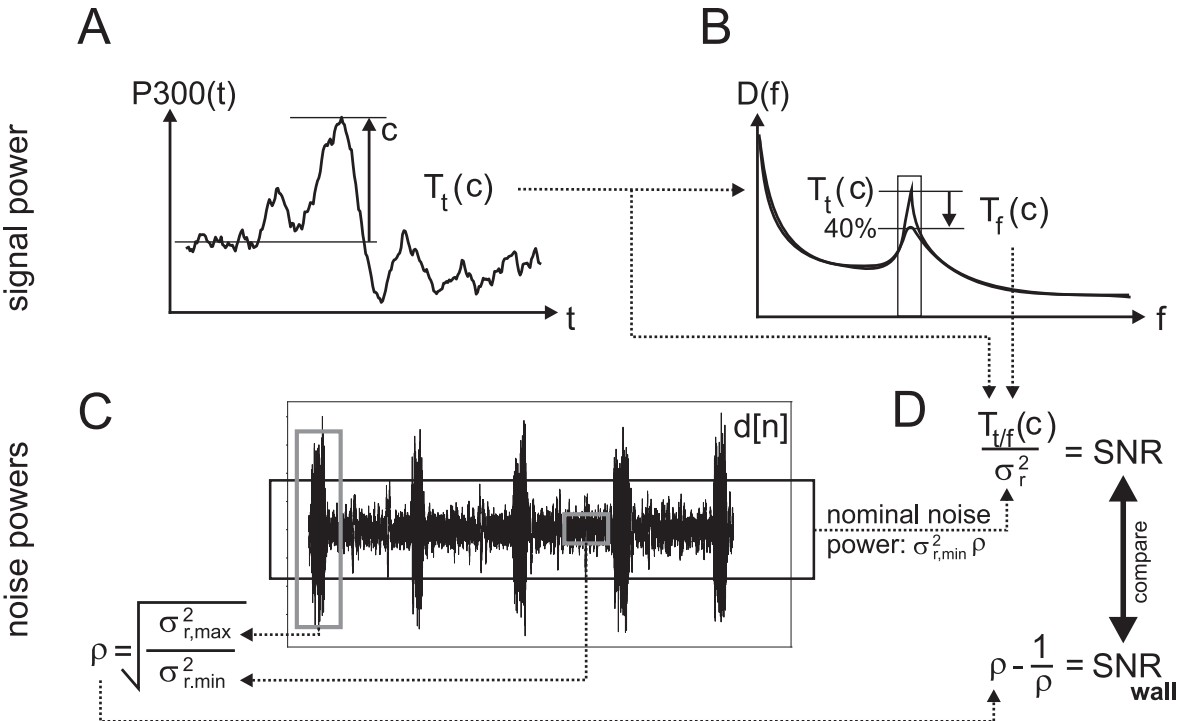

**Fig 2. Flow diagram illustrating the calculation of the SNR and SNR-wall.** A) Time domain signal for the SNR calculation takes the power $T_t(c)$ of the P300 peak as the consciously generated EEG change. B) Frequency domain signal power for the SNR calculation takes the power $T_t(c)$ and assumes a conscious 40% reduction of the signal power in a narrow band around a peak. C) For the overall SNR the nominal variance $\sigma_r^2$ is calculated over the whole EEG recording. To calculate the SNR-wall the maximum $\sigma_{r,max}^2$ variance and minimum $\sigma_{r,min}^2$ variance is detected with a sliding window sample by sample. D) If the SNR is greater than the SNR-wall then a conscious change in the EEG can be detected. If the SNR is less than the SNR-wall then it is impossible to detect a conscious change.

- **Time-domain power of the signal**: The P300 peak can be used straight away for a time domain calculation of its power (Fig 2A):

$$T_t(c) = c_{max}^2 \qquad (27)$$

where the $T_t(c)$ is the power of the pure EEG in the time domain.

- **Frequency-domain power of the signal**: BCI systems using the frequency domain change the power of a narrow frequency band (Fig 2B), for example by motor imagination. Here, we assume that:

  - The EEG power generated in a narrow frequency band is comparable to the power generated in the time domain $T_t(c)$. This is indicated between the panels Fig 2A and 2B by noting that the peak powers are identical.

  - However, because motor imagination *reduces* power consciously the actual conscious power $T_f(c)$ is only as strong as the *reduction* of power. Here, we assume a 40% reduction of power.

Given that only the reduction is the conscious power change *c* and anything else will be absorbed in the noise term *r* we can calculate the pure signal power (in contrast to the noise)

as:

$$T_f(c) = T_t(c) - T_t(c) \cdot 40\% \tag{28}$$

After having calculated the power of the signal $T_f(c)$ we now need to calculate the power of the noise. Given that the small P300 evoked potentials $c$ are buried in the EEG $d$, one can assume that the noise variance containing a consciously generated signal and one without are basically identical [14]: $\sigma_r^2 \approx \sigma_d^2$. Thus we take the variance of the EEG epoch $d[n]$ (Fig 2D) as

$$\sigma_r^2 \approx T(d) \tag{29}$$

where the power of the EEG epoch $T(d)$ is calculated with Eq 7 which is the average power or variance.

**Step 3: Comparing SNR and SNR-wall: Determining if conscious control can be detected.** If the SNR of the EEG (Eq 10) is above the SNR-wall (Eq 20) then detection of the conscious EEG change $c$ is possible. Otherwise not.

For every task, for example "Sudoku", there will be individual pairs of SNR and SNR-wall values from every subject. Because the SNR and SNR-wall values are calculated over all subjects they are random variables. A t-test is used to determine if the SNR for all subjects is significantly above the SNR-wall for each task and each of the four post-processing scenarios.

## Results

We are going to describe the results in three steps:

1. **Investigating the raw data**: The different datasets (jaw clench, lying eyes closed/open, word search, Sudoku, phone app, reading, colouring) used in this study exhibit different signal and noise characteristics in their time- and frequency domain. We will use subject #20 as a *representative example* to point out the differences between the datasets.

2. **Walk-through of the calculation of the SNR and SNR-wall of one subject for both jaw clench and reading**: As instructional examples we calculate step by step the SNRs and SNR-walls of *one* subject for *jaw-clench* and *reading* at different EEG frequency ranges such as: full range, only low frequencies (0.1–3 Hz), wide bandpass (8–18 Hz), narrow bandpass (8–12 Hz) and derivative.

3. **Statistical analysis with t-test**: Using the same approach as in step 2 but for *all* subjects enables us to do a statistical analysis with a t-test which determines if the SNR of a task is *significantly* larger than the SNR-wall and thus conscious EEG changes can significantly be detected.

### Investigating the raw data

Fig 3 shows the measurement results taken from subject 20 which are representative of the artefacts encountered during the different experimental tasks which are listed on the left-hand side: jaw clench, lying eyes closed/open, word search, Sudoku, phone app, reading, colouring. Fig 3A shows the time domain plots for 60 seconds. The signals plotted have only the DC and 50 Hz main interference removed and otherwise represent the full range of raw signals. Panel Fig 3B shows the power spectra in logarithmic units of the corresponding time domain plot to the left in Fig 3A. The dotted line at 20 Hz marks the point from where muscle noise

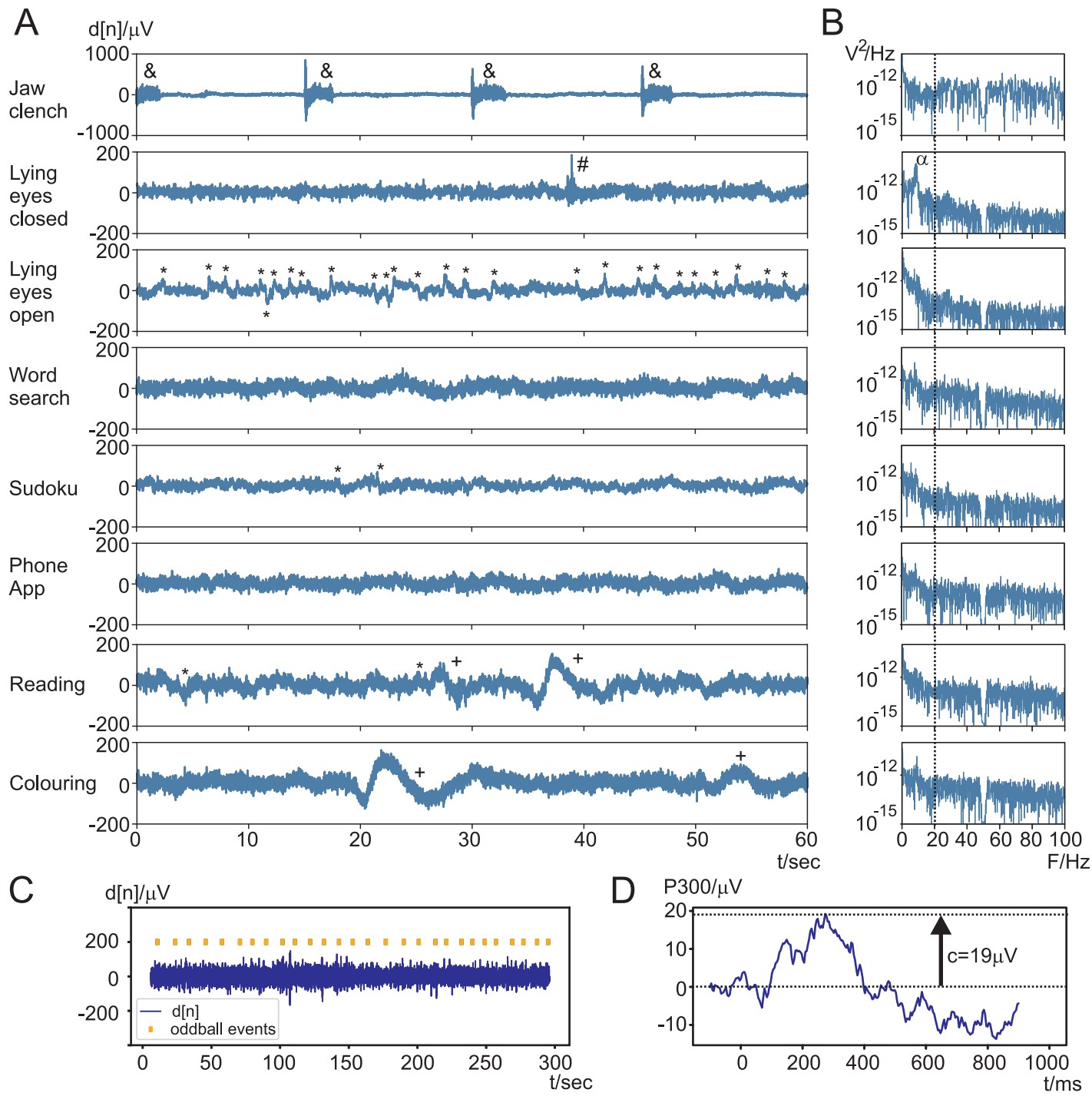

**Fig 3. Complete dataset of a representative recording (subject 20).** A) Time domain of the different recordings ($d[n]$ in $\mu V$ against $s$). B) Power spectra of $d[n]$ of the corresponding recordings ($V^2$/Hz). C) Raw P300 recording ($\mu V$ against $s$). D) Averaged P300 response ($\mu V$ against $ms$).

contribution will decrease towards lower frequencies [15] or in other words: we will expect muscle noise for frequencies above 20 Hz but also to a lower degree down to 10 Hz. We are now going through the different tasks step by step. We start with "jaw clenching" followed by "lying down, eyes closed" being the tasks with the largest difference in noise amplitudes where

jaw clenching has about 50 times more noise than lying down. Note that, for the noise wall calculation, it is the *ratio* between the highest and lowest noise variances that are used (Eq 25), not the absolute amplitudes. The ratio is again highest for the jaw clench:

- **Jaw clenching**: For this task the subject had to clench their jaw every 15 seconds when indicated by the researcher and these have been marked with an "&" in Fig 3A. Peak amplitudes at the onset of the jaw clench reach nearly 1 mV while in the pauses the amplitude is just around 20 $\mu$V. Its corresponding spectrum in Fig 3B is almost flat over the whole frequency range. Compared to the relaxed state of lying down the spectral power during jaw clenching is approx 100 times larger for frequencies above 20 Hz. This experiment is useful to gain an insight into which spectral components are affected by muscle noise and confirms that in particular frequencies above 20 Hz are affected but this is of course no hard cutoff but the influence of muscle noise towards lower frequencies decreases gradually.

- **Lying down, eyes closed**: Here, the subject was asked to relax as much as possible and since the eyes were closed there are no eyeblink artefacts. The amplitude is at about 20 $\mu$V with small fluctuations. However, muscle activity can still occur as marked with "#" where the subject had to swallow and the muscle activity can clearly be seen. In the frequency spectrum an alpha peak at 10 Hz emerges which is expected when closing the eyes.

- **Lying down, eyes open**: With the eyes open the alpha peak vanished and the subject was looking at different locations at the ceiling and in the lab which shows up as a mix of eye-blink and eye-movement artefacts marked with "*". Generally subjects tended to look around frequently as it is a slightly unusual situation.

- **Word search**: Word search engages the eyes but only at a narrow viewing angle looking at a sheet of paper which causes only small eye saccades. However, the spectrum between 20–50 Hz is elevated indicating muscle activity, possibly because of tense muscles during the task.

- **Sudoku** appears to have similar muscle activity as lying down with eyes closed, with little facial muscle tone. However, since the eyes were open and the subject had to scan the Sudoku grid a few sudden jumps in the EEG potential can be identified which were saccades indicated with a "*".

- **Phone app game**: The phone app has a similar behaviour as "word search" in that the power spectrum is elevated 20–50 Hz suggesting again high facial muscle activity.

- **Reading aloud** creates a mix of artefacts as various facial muscles are needed for spoken word and the eyes need to scan the text which cause eye artefacts as well. The frequency spectrum above 20 Hz in Fig 3B is elevated and stays elevated up to 100 Hz. In addition lower frequency components from eye-blinks and saccades "*" appear towards 0.1 . . . 3 Hz. Very low frequency components indicated at "+" were generated because the subject was turning the page and then touching their cheek while reading.

- **Colouring**: Here, the subject had to take different pens and colour a line drawing. This task is an activity where the colouring makes the whole body move in the rhythm of the strokes made with the pen. Interestingly these pen-strokes were in the region of 10 Hz and created a weak fake alpha peak in the spectrum. In addition the activity also created larger movement artefacts at "+" where the subject switched pens.

   The final two panels in Fig 3C and 3D show the recording of the P300 experiment. This experiment determines the *signal power* for the SNR calculation (see Eq 27). Recall that it is

not ethical to paralyse a healthy subject to determine their pure EEG signal power and for that reason we use the P300 oddball reaction of the brain to have a conscious voltage change to a surprising stimulus. The subject is presented with the oddball event at irregular intervals as indicated in Fig 3c as "oddball events". The actual change in EEG is small and buried in the original electrode signal $d[n]$. After event triggered averaging we arrive at Fig 3D where the oddball event happened at $t = 0$ ms and the EEG response then reached its peak of $c_{max} = 19 \mu V$ after 300 ms.

## Walk-through of the calculation of the SNR and SNR-wall of one subject for both jaw clench and reading

We now present a walk-through of how to calculate the SNR and the SNR-wall for the "jaw clench" task. This serves as an instructional example for the statistical analysis for all subjects further below. Fig 4 shows the time- and frequency domain signals during jaw clenching in panels A-M and B-N, respectively. Panels A and B are identical to the 1st row in Fig 3. The column C-O shows the calculated SNR-wall and SNR values based on the time domain data of the corresponding panels A-M. Let us first focus on the first row Fig 4A–4C and show step by step how to calculate the SNR-wall and SNR for it. We start with the SNR-wall calculation (Eqs 14 and 20 & Fig 2C and 2D) for which we need the minimum variance (here: $\sigma_{r,min}^2 = 1.15 10^{-10} V$) and maximum variance (here: $\sigma_{r,max}^2 = 2.00 10^{-8} V$). This results in $\rho = 13.21$ and an SNR-wall of $SNR_{wall} = 11.18$ dB which is shown in Fig 4C. This means that the SNR of the electrode signal needs to be larger than $SNR_{wall} = 11.18$ dB to be able to detect conscious changes in the EEG. Consequently as a next step we are going to calculate the SNR of the electrode signal. The overall SNR of the full range electrode signal is the ratio of the amplitude of the conscious EEG change $c_{max}$ against the nominal noise power (see Eq 26). The conscious EEG change was already determined in Fig 3 as $c_{max} = 19 \mu V$ and its power $T_t(c) = c_{max}^2 = 3.70 10^{-10} V^2$. The nominal noise power $\sigma_r^2$ can either be calculated with $\sigma_r^2 = \rho \sigma_{r,min}^2$ or directly from the electrode signal $d[n]$ (Eq 26) and results here in: $\sigma_r^2 = 2.75^{-09} V^2$. This leads to an SNR of $-8.71$ dB which is shown in Fig 4C. Since the SNR is substantially below the SNR-wall, detection of conscious changes of EEG are *not* possible in this case. This comes as no surprise as the SNR-wall is determined by the *ratio* between the minimum noise variance $\sigma_{r,min}^2$ and the maximum one $\sigma_{r,max}^2$. The wider the distance between these two lines in Fig 4A the higher the SNR-wall.

However, filtering of the electrode signal changes both the SNR-wall and the SNR which might allow detection of conscious changes. We are now discussing different filtering approaches. The calculation of both the SNR and SNR-wall is identical to the methodology described above with the exception that we assume that subjects can only *reduce* consciously an EEG frequency band by a certain percentage ("desynchronisation") which is set to 40% thus the signal power for the SNR calculation is set to $T_f(c) = (60\% \cdot 19 \mu V)^2$. We are now showing the impact of filtering on the SNR-wall and SNR:

- 0.1–3 Hz, Fig 4D–4F: This low frequency bandpass filter covers the frequency range of eyeblinks and saccades. While in the original trace Fig 4A these have not been apparent, now they are easily identifiable in Fig 4D. In particular during the first 10 seconds the subject blinked repeatedly which has been marked with the "*". These artefacts drive up the difference between chunks of small noise power and large noise power which in turn increase the SNR-wall. However, also the jaw muscle artefacts show up because of their impulse-like shape they trigger large impulse responses of the 0.1–3 Hz bandpass filter which increase the maximum variance $\sigma_{r,max}^2$ even further. In contrast the SNR stays again negative at $-6$ dB

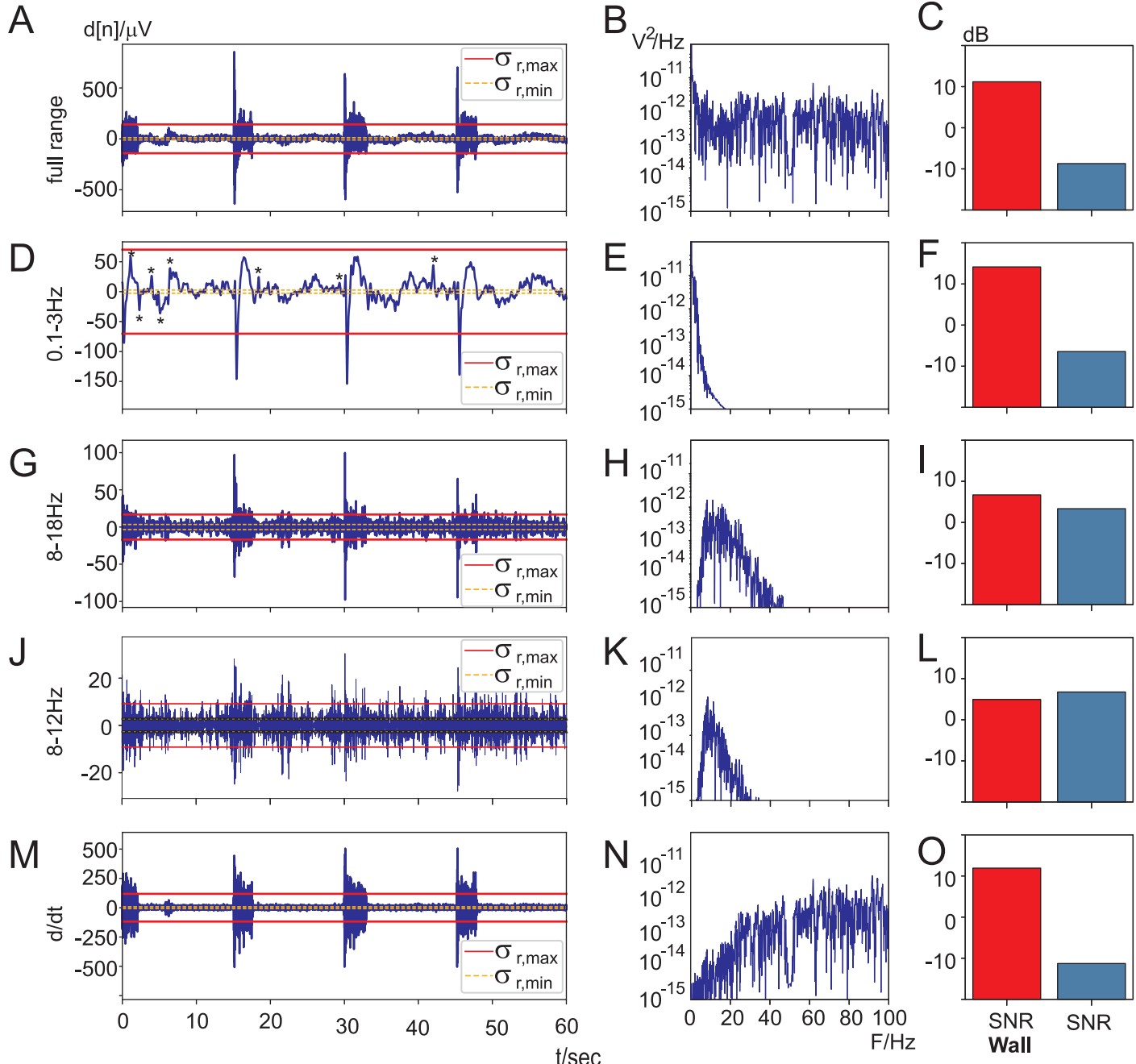

**Fig 4. SNR and SNR-wall calculations for the "jaw clench" task.** Columns A-M: electrode signals in the time domain with lines for the maximum variance $\sigma_{r,max}^2$ and minimum variance $\sigma_{r,min}^2$ while using a sliding window of 2 sec. Columns B-N: frequency power spectra of the corresponding time domain signals of columns A-M. Columns C-O: SNR-wall and SNR calculations of the corresponding time domain signals of columns A-M.

while the noise wall is driven up to + 14$dB$. Consequently, it is not possible to detect conscious changes in EEG.

- 8–18 Hz, Fig 4G–4I: This moderately narrow bandpass filter is a popular choice in BCI applications [11] where subjects alter consciously the amplitudes of frequencies in this band. The

ratio between lowest $\sigma^2_{r,min}$ noise- and highest $\sigma^2_{r,max}$ noise-power is comparable to Fig 4A but the SNR has improved due to less muscle activity from the jaw muscles which is only prominent above 20 Hz. Still the SNR lies below the SNR-wall and thus detection of conscious EEG changes are not possible.

- 8–12 Hz, Fig 4J–4L: This bandpass filtered signal stays substantially below the 20 Hz mark which promises less muscle activity. Indeed, the effects of the jaw clenches are clearly strongly suppressed in Fig 4J. This reduces the noise wall to 5$dB$ and improves the SNR to 7$dB$ which means that it is possible to detect conscious alpha frequency changes even with the jaw muscles being active.

- $d/dt$, Fig 4M–4O: Applying the derivative to the electrode signal is a popular choice in BCI as well [7]. The derivative or perhaps higher order derivatives act as a highpass filter which removes DC and lower frequencies such as eye-movements or eye blinks but favours the EMG frequency band. Indeed, when comparing the full range electrode signal in Fig 4A with the highpass filtered one in Fig 4M the jaw muscle activity is strongly emphasised. This leads to a high SNR-wall of 12$dB$ and a low SNR at −11$dB$ and making it impossible to detect any conscious EEG changes in the presence of strong non-stationary muscle activity.

While the jaw clench is excellent for demonstrating the effect of strong non-stationary muscle noise the reading task is more realistic in that it contains non-stationary muscle activity at various levels, eye movements and also movement artefacts. Fig 5 follows the same layout and the same parameters as Fig 4. The full range signal (Fig 5A–5C) again exhibits a large difference between the lowest noise power $\sigma^2_{r,min}$ and the highest noise power $\sigma^2_{r,max}$ which leads to a high SNR-wall of 7 dB and at an SNR of −4 dB it is not possible to detect conscious EEG changes here. We now move on to the filtered electrode signals (Fig 5D–5O). The filtering of the electrode signal between 0.1–3 Hz emphasises eye movements and movement artefacts and shows numerous saccade like voltage changes which are characteristic of reading where the eyes need to scan the page. The large spike around 40 sec is a movement artefact. Together these artefacts cause a large difference between the smallest noise power and maximum noise power which in turn leads to a noise wall which is again much higher than the SNR (Fig 5F). The frequency range of 8–18 Hz is the one used in motor imagination and rejects most of the muscle noise. This results in a lower SNR-wall and higher SNR (Fig 5I) so that now detection of conscious EEG changes are possible. Similarly, the frequency range of 8–12 Hz for alpha waves allows detection of conscious EEG changes (Fig 5L). A curious case is the use of the derivative (Fig 5M–5O) which has been detrimental during the jaw muscle contractions but here leads to a low SNR-wall and an SNR which is just above the SNR-wall, potentially allowing the detection of conscious changes of EEG. This could have been of course just chance and different for the other subjects which illustrates that both SNR and SNR-wall are *random variables* where every subject in an experiment generates individual SNR-wall & SNR pairs. This calls now for a statistical evaluation with all subjects and a t-test which tests for the SNR being significantly above the SNR-wall which is described below.

## Statistical analysis

Fig 6 shows the statistical evaluation of the SNR-walls against the SNRs. Each panel Fig 6A–6E shows the SNR-walls and SNRs for the different tasks ranging from lying down to jaw clenching. Panels A-E themselves represent different filtering methods of the electrodes signals. Both SNR and SNR-walls are shown in dB on a scale from -15 dB to +15 dB. The y-axis shows the results for each separate task so that one can decide if it is possible to detect a conscious change in the EEG reliably or not. Both SNR and SNR-walls are random variables from the different

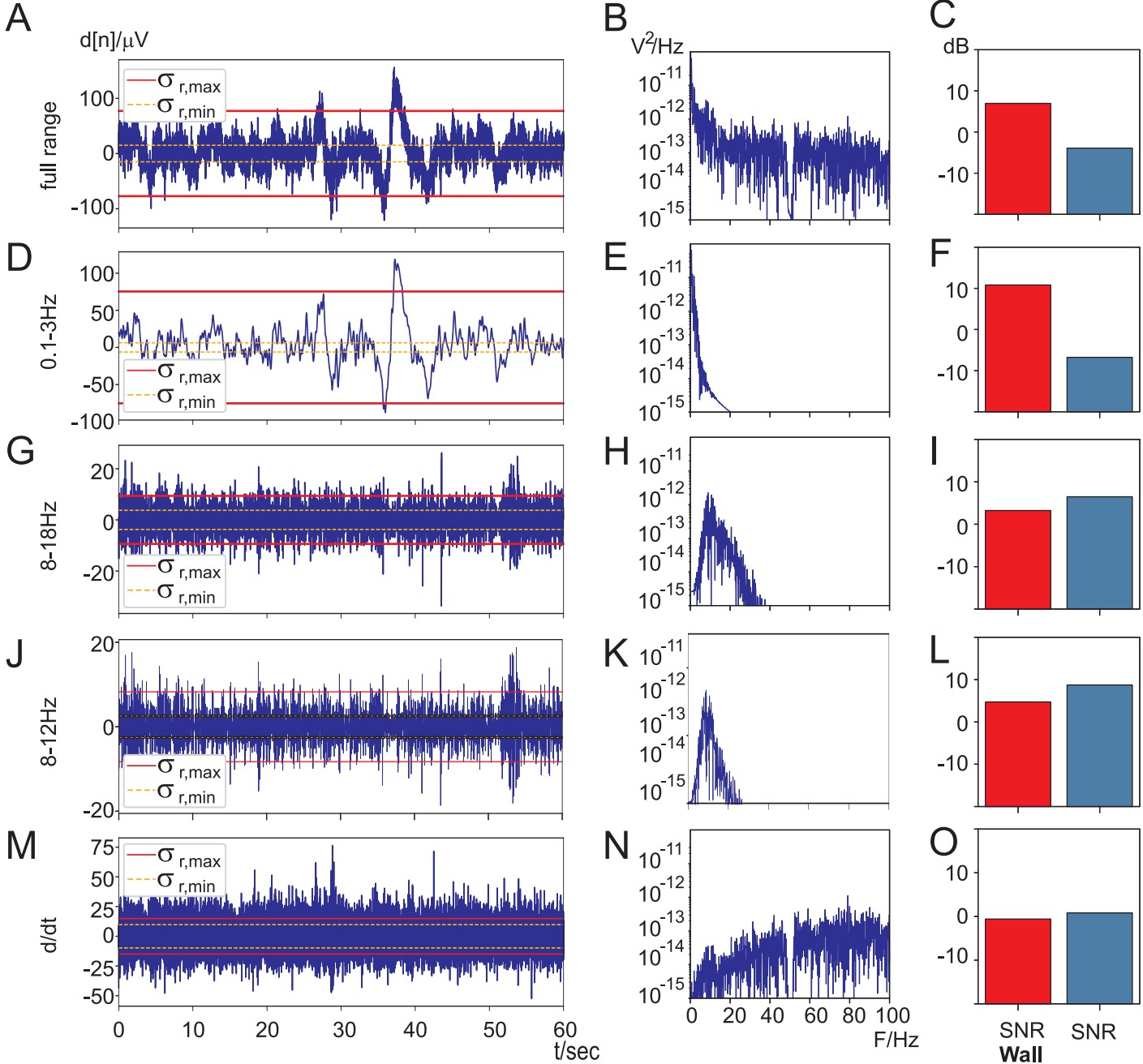

**Fig 5. SNR and SNR-wall calculations for the "reading" task.** Columns A-M: electrode signals in the time domain with lines for the maximum variance $\sigma^2_{r,max}$ and minimum variance $\sigma^2_{r,min}$ while using a sliding window of 2 sec. Columns B-N: frequency power spectra of the corresponding time domain signals of columns A-M. Columns C-O: SNR-wall and SNR calculations of the corresponding time domain signals of columns A-M.

participants and a t-test ($p < 0.05$) was used to determine if the SNR is significantly above the SNR-wall which then predicts that conscious EEG changes can be detected for a specific experiment using a specific causal filtering technique. We describe the statistical analysis first against the different pre-processing methods and then in relation to the different tasks.

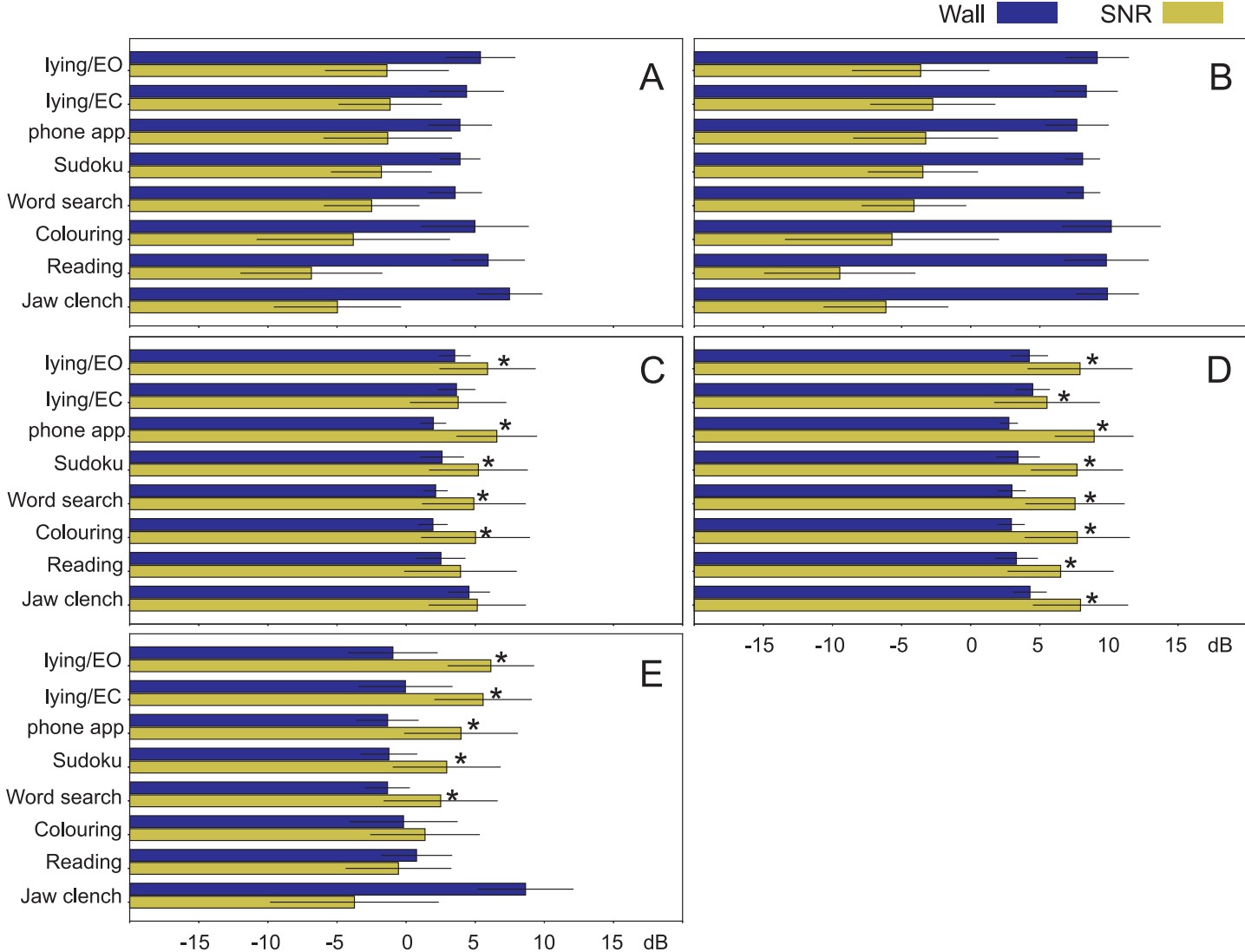

**Fig 6. Statistical results.** SNR versus SNR-wall for A) full range electrode signal with only highpass filtering at 0.1 Hz and 50 Hz removal, B) 0.1–3 Hz bandpass filtered, C) 8–18 Hz bandpass filtered, D) narrow bandpass 10±2 Hz and E) derivative. The bars show the average for the SNR-wall and the SNR with error bars for standard deviation. A t-test was used to determine if conscious EEG changes can be detected at $p < 0.05$ and the "*" identifies the experimental and filtering conditions where this is significantly possible.

**A)** 0.1 > Hz: Fig 6A shows the SNR/SNR-wall pairs after only highpass filtering above 0.1 Hz and mains removal. The task "jaw clench" has an average SNR-wall of 8 dB but given that the SNR is at −5 dB it is significantly not possible to detect conscious EEG here. Generally in Fig 6A all average SNR values are lower than the average SNR-wall values and the corresponding t-test reports that electrode signals after minimal filtering significantly cannot detect a conscious change of EEG at all.

**B)** 0.1–3 Hz: As discussed in the previous sections this frequency range strongly emphasises eye movements and movement artefacts which leads to high average SNR-walls and very low SNR values for all tasks. Consequently the t-test provides again no significant case where conscious EEG changes could be detected.

**C**) 8–18 Hz: This bandpass filter yields significant results predicting the detection of conscious EEG changes for: lying (eyes closed), using a phone app, playing Sudoku, doing word search and colouring. Reading which engages a large amount of facial muscles and the jaw clench as a worst-case scenario obliterate any detection effort. It is interesting to see that lying down with eyes closed won't allow detection which anecdotally points to the subjects being more tense having a higher muscle activity.

**D**) 8–12 Hz: Best detection ability is achieved with a narrow frequency bandpass around the alpha band with all tasks allowing detection of a consciously changed EEG.

**E**) $d/dt$: Here, pre-filtering is done by a derivative which is popular for realtime BCI [7]. Here, lying down with eyes closed and open, using a phone app, playing Sudoku, doing word search and colouring allow the detection of a conscious EEG change. Colouring, reading and jaw clench have the highest EMG components and fail the t-test which means that conscious EEG changes cannot be detected. As already observed earlier the derivative emphasises EMG activity which is apparent in panel E for jaw clench where the average SNR is about 10 dB lower than the noise wall and creates a clear cut non-significant result.

It is interesting to note that with the derivative keeping in particular EMG intact that there is a clear hierarchy of SNR values from eyes closed having the highest SNR down to the jaw clench for the lowest. In contrast, the bandpass filter approaches in panels C) and D) do not have this dependence with both bandpass ranges below the 20 Hz boundary for EMG confirming again that keeping BCI detection below 20 Hz is the safest bet to avoid EMG interference.

In the previous section we have ordered the results according to the *pre-filtering* of the electrode signal. Of equal interest is which *task* allows the detection of conscious EEG changes and which renders it impossible to do so. For this purpose we have re-ordered the results of Fig 6 and show them task by task in Table 1. As in Fig 6 the "*" in Table 1 indicate that conscious EEG changes can be significantly detected. The labels A-E match the panel labelling in Fig 6 and are the different pre-filtering methods.

**Lying down, eyes open**: Conscious detection of EEG is possible here for moderate wideband prefiltering (8–18 Hz), narrow alpha band filtering (10±2 Hz) or using the derivative which acts as a highpass. Lying flat down and having the eyes open mainly creates eye blink artefacts which, without any filtering, cause a high ratio between the minimum and maximum noise variances which in turn result in a high SNR wall. EEG detection is possible if these eye-blink artefacts are removed by a highpass filter characteristic.

**Table 1. Detection of conscious EEG changes significantly possible ordered by task.**

|  | A 0.1 > Hz | B 0.1–3 Hz | C 8–18 Hz | D 10 ± 2 Hz | E $d/dt$ |
|---|---|---|---|---|---|
| lying/EO |  |  | * | * | * |
| lying/EC |  |  |  | * | * |
| phone app |  |  | * | * | * |
| Sudoku |  |  | * | * | * |
| Word Search |  |  | * | * | * |
| Colouring |  |  | * | * |  |
| Reading |  |  |  | * |  |
| Jaw clench |  |  |  | * |  |

**Lying down, eyes closed**: With lying down and eyes closed detection of conscious EEG changes is still possible when focusing on the narrow alpha band or using the derivative as a highpass filter. However wider bandpass filtering yields no significant results. As mentioned above closing the eyes in a lab is slightly uncomfortable which seems to result in a higher muscle tone. Comparing Fig 6c between eyes open and closed only the SNR changes but not the SNR-wall which means that the actual ratio of min and max noise variance is not different between those two tasks but rather the constant muscle tone which is reflected in a higher SNR while having the eyes closed.

**Phone app, Sudoku, Word search**: As outlined further above these three tasks have only moderately elevated non-stationary muscle activity and eye-movements. The muscle activity mostly arises from being a bit more tense when solving the tasks but causes only low ratios between highest noise variance and lowest noise variance which in turn results in low SNR-walls. Even the simple derivative which favours muscle noise still yields significant results. Again, the low frequency filtering and wideband filtering make it impossible to detect conscious EEG changes because the low frequency eye movements cause high SNR-walls.

**Colouring**: This is now a task which involves strong arm muscle activity but also most likely compensatory muscle activity in the entire body such as neck muscles. This results in higher muscle noise levels compared to the previous tasks and now the filtering with the derivative no longer allows detection of conscious EEG changes. However, wideband prefiltering (8–18 Hz) and narrow alpha band filtering (10±2 Hz) still allow significant EEG detections.

**Reading aloud**: This task causes irregular contraction of facial muscles and the eyes need to scan the text which cause eye artefacts as well. This non-stationary noise leads to high SNR walls which prevent the detection of conscious EEG changes except when performing narrow band filtering around the alpha band. This filtering steers clear of both eyeblink artefacts and higher frequency muscle noise as much as possible.

**Jaw clenching**: The jaw muscles create very high amplitude bursts of muscle activity which are about 50 times higher than the baseline noise. Without any filtering this results in high ratios of minimum and maximum noise variances which in turn results in very high noise walls. Similarly to reading aloud removing the non-stationary muscle noise is crucial to be able to detect conscious EEG changes and is only possible if detection focuses just on the narrow alpha band.

Overall tasks with strong non-stationary facial muscle activity such as reading aloud or jaw clenching allow only significant detection of conscious EEG changes if one performs narrow band filtering around the alpha band which steers clear as much as possible of both muscle noise and eye movement artefacts. However, eye movement artefacts are less of a problem than muscle artefacts because their frequencies do not overlap with the standard BCI frequency detection ranges. Those tasks which contain mostly eye movements but little muscle noise such as playing a video game allow detection of conscious EEG changes with a wideband BCI bandpass (8–18 Hz) but also with a derivative which only attenuates eye-movements but not muscle noise.

## Discussion

In this paper we have introduced an objective hard criterion which determines if it is possible to detect conscious EEG changes in a recording contaminated with non-stationary noise. Specifically, this methodology requires the SNR of an electrode signal to be higher than its SNR-

wall so that conscious changes in EEG can be detected. The SNR-wall is calculated by taking the ratio between the minimum and maximum noise variances of the electrode signal (Eq 20). The SNR is calculated using the standard definition of SNR which is here the ratio between the consciously generated EEG component against the nominal noise level (Eq 10). Both equations result from sound analytical calculations and were then used to calculate all SNR-wall and SNR values for all experimental conditions and tasks. In other words, no manual selections of the SNR-walls, SNRs or thresholds are needed for the BCI-walls framework protecting it from subjective, possibly arbitrary choices. We then determined the SNR-wall for a range of different tasks with non-stationary noise while applying different filtering techniques. Wideband filtering and filtering in the delta band make it impossible to detect any conscious EEG changes. On the other hand filtering around the alpha band allows EEG detection even under high non-stationary muscle interference during jaw clenching.

As outlined in the introduction, EEG is contaminated with different forms of noise where EMG is the hardest to remove because of its broad frequency spectrum [2] ranging from 20 to 80 Hz. The gold standard to measure the EMG contribution is by neuromuscular blockade where a subject is temporally paralysed and, thus, generates zero EMG. Measuring brain activity from a paralysed subject reveals substantial EMG contamination above 20 Hz [15] and matches our results where the SNR-wall is lowest if one uses a narrow bandpass around the 10 Hz alpha band [12] steering clear of the EMG spectrum.

An existing method of how to determine if EMG interference is detrimental to EEG detection is to use the amplitude of a bandpass filtered electrode signal and suspend data collection when it is above a certain threshold [2]. Another suggested method is to use a montage consisting of a large number of electrodes and then apply independent component analysis (ICA) to the electrode signals [2, 3, 16] which separates noise from EEG which then allows the calculation of the SNR. The ICA could also be used to determine the SNR for the BCI-wall methodology. Instead, we have chosen the P300 methodology to determine the pure consciously controlled EEG power because ICA requires by definition more than one electrode and we recorded from just one. Future research could also calibrate the SNRs obtained from the ICA with those from the P300 responses. Be it P300 or ICA they are in stark contrast to determining the SNR by paralysing a participant temporarily [15, 17] which would certainly yield the best results but no doubt would raise substantial ethical concerns.

So far ICA has only been suggested to determine the SNR but it can also be used to determine the SNR-wall because the minimum $\sigma_{r,\min}^2$ and maximum noise variances $\sigma_{r,\max}^2$ are readily available in the independent component(s) of the ICA carrying the noise. Since the ICA is a well-established standard tool for EEG offline analysis calculating both the SNR and SNR-wall could just be a simple additional step which then allows the robust evaluation of an experiment.

However, it appears that most BCI studies and reviews [18, 19] stay silent about how they have dealt with noise interference. The first comprehensive literature survey investigating artefact removal [20] finds that most BCI papers do not report whether or not they have considered the presence of EMG (67.6%) or EOG (53.7%) artefacts in brain signals. This situation has not improved much ten years later [21] where 41% of the studies did not mention any artefact removal process of EEG and even where artefacts were mentioned 22% of those studies did not do any cleaning or artefact removal. Given that the BCI-walls methodology has been proven to be analytically sound [8] and requires only standard EEG recording parameters such as signal- and noise-variances it should be part of a standard process to evaluate whether reliable EEG detection is possible or not. Pre-registrations of experimental designs could also specify a BCI-wall test which could then later provide a robust decision point as to whether

EEG recordings have been sufficiently high quality to allow the detection of conscious EEG changes or if the data needs to be discarded.

SNR-wall theory was developed in the field of telecommunications [8] where it is common that a multitude of transmitters create non-stationary background noise [22–27] and, for example, making it impossible for a mobile phone to communicate with its base station with an increasing number of transmitters. The SNR-wall calculations are able to make strong predictions about the capability of a telecommunications system to work as intended, and similarly this should be also standard practise for BCI systems.

## Acknowledgments

We thank Nicholas J Bailey for his constructive feedback on the manuscript.

## Author Contributions

**Conceptualization:** Bernd Porr.

**Data curation:** Lucía Muñoz Bohollo.

**Formal analysis:** Bernd Porr, Lucía Muñoz Bohollo.

**Investigation:** Lucía Muñoz Bohollo.

**Methodology:** Bernd Porr.

**Software:** Bernd Porr, Lucía Muñoz Bohollo.

**Supervision:** Bernd Porr.

**Writing – original draft:** Bernd Porr, Lucía Muñoz Bohollo.

**Writing – review & editing:** Bernd Porr, Lucía Muñoz Bohollo.

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
