## [Decision Letter · Decision Letter 0]

17 Mar 2023

PONE-D-22-30139BCI-Walls: A robust methodology to predict success or failure in brain computer interfacesPLOS ONE

Dear Dr. Porr,

Thank you for submitting your manuscript to PLOS ONE. After careful consideration, we feel that it has merit but does not fully meet PLOS ONE’s publication criteria as it currently stands. Therefore, we invite you to submit a revised version of the manuscript that addresses the points raised during the review process.

ACADEMIC EDITOR: The reviewers see potential in the study and it may be suitable for publication in this journal after major revisions as suggested by the reviewers. 

We look forward to receiving your revised manuscript.

Kind regards,

Noman Naseer, PhD

Academic Editor

PLOS ONE

Journal Requirements:

a) Did participants provide their written or verbal informed consent to participate in this study?

b) If consent was verbal, please explain i) why written consent was not obtained, ii) how you documented participant consent, and iii) whether the ethics committees/IRB approved this consent procedure

Additional Editor Comments:

Reviewer's see potential in the paper. However, major revisions are reuqired before the paper is ready for publication,

Reviewers' comments:

Reviewer's Responses to Questions

**Comments to the Author**

1. Is the manuscript technically sound, and do the data support the conclusions?

Reviewer #1: Yes

Reviewer #2: Yes

2. Has the statistical analysis been performed appropriately and rigorously? 

Reviewer #1: No

Reviewer #2: No

3. Have the authors made all data underlying the findings in their manuscript fully available?

Reviewer #1: Yes

Reviewer #2: Yes

4. Is the manuscript presented in an intelligible fashion and written in standard English?

Reviewer #1: Yes

Reviewer #2: Yes

5. Review Comments to the Author

Reviewer #1: The authors presented an interesting aspect, an objective hard criterion which determines if it is possible to detect conscious signal in an EEG at all in a recording contaminated with non-stationary noise, which they call BCI-wall. However, I have some concerns related to the evaluation and test parts.

The abstract does not reflect the work of the authors. It must be rewritten again. It is hard to understand the scientific and technical parts to calculate SNR-wall and why it can be considered as hard and measurable criterion to determine if a BCI experiment can detect conscious EEG changes or not.

Applying ICA might be another way to evaluate the results of the proposed techniques. In addition, the authors didn't explain the impact of the eye movements (especially 0.1-3Hz) too on SNR-wall.

The authors must discus more about statistical evaluation techniques to validate their BCI-wall aspect in several EEG datasets.

Reviewer #2: The paper proposes a novel methodology, called BCI-Walls, to predict the success or failure of Brain-Computer Interfaces (BCIs) based on the signal-to-noise ratio (SNR) of the electroencephalogram (EEG) data. The authors tested this methodology on a dataset of 20 participants and showed promising results.

However, the paper has some limitations that need to be addressed. Firstly, the results shown in Figure 3 are not clearly explained. Specifically, the tasks involving jaw clenching and reading, which require movement of the facial muscles, are not elaborated. The authors should provide more details on how these tasks were performed and how they affect the SNR of the EEG data.

Secondly, while the authors provided some evidence to support the validity of the BCI-Walls methodology, the results are not conclusive. The authors did not provide a clear picture of how the SNR-wall method supports their proposed methodology. They should provide more detailed analysis and explanation of the results to support the efficacy of their approach. Additionally, the authors should provide further validation of their results by applying a statistical test such as t-test to assess the significance of their findings.

Despite these limitations, the paper presents a promising methodology for predicting the success or failure of BCIs. The authors should address the limitations mentioned above to improve the clarity and validity of their findings. Moreover, the paper could benefit from a more detailed comparison with existing methods to demonstrate the advantages of the proposed methodology.

6. PLOS authors have the option to publish the peer review history of their article (what does this mean?). If published, this will include your full peer review and any attached files.

Reviewer #1: No

Reviewer #2: No

---

## [Author Response · Author response to Decision Letter 0]

28 Apr 2023

See the uploaded PDF labelled "response to the reviewers" which is called loc.pdf.

---

## [Decision Letter · Decision Letter 1]

24 May 2023

PONE-D-22-30139R1BCI-Walls: A robust methodology to predict if conscious EEG changes can be detected in the presence of artefactsPLOS ONE

Dear Dr. Porr,

Thank you for submitting your manuscript to PLOS ONE. After careful consideration, we feel that it has merit but does not fully meet PLOS ONE’s publication criteria as it currently stands. Therefore, we invite you to submit a revised version of the manuscript that addresses the points raised during the review process.

We look forward to receiving your revised manuscript.

Kind regards,

Noman Naseer, PhD

Academic Editor

PLOS ONE

Journal Requirements:

Additional Editor Comments (if provided):

Most of the comments have been addressed. Few minor comments still require attention.

Reviewers' comments:

Reviewer's Responses to Questions

**Comments to the Author**

1. If the authors have adequately addressed your comments raised in a previous round of review and you feel that this manuscript is now acceptable for publication, you may indicate that here to bypass the “Comments to the Author” section, enter your conflict of interest statement in the “Confidential to Editor” section, and submit your "Accept" recommendation.

Reviewer #2: All comments have been addressed

2. Is the manuscript technically sound, and do the data support the conclusions?

Reviewer #2: Partly

3. Has the statistical analysis been performed appropriately and rigorously? 

Reviewer #2: Yes

4. Have the authors made all data underlying the findings in their manuscript fully available?

Reviewer #2: Yes

5. Is the manuscript presented in an intelligible fashion and written in standard English?

Reviewer #2: Yes

6. Review Comments to the Author

Reviewer #2: The proposed methodology is a significant contribution to the field of BCI research and has the potential to improve the accuracy and reliability of BCIs for a range of applications.

However, there are a few areas that could be improved. The paper does not provide a clear explanation for how the SNR-wall threshold was selected, which could be a limitation. To enhance the reliability of the results, it would be beneficial to separately analyze each task in the experimental setup rather than combining all tasks together. Additionally, while the paper proposes a validation method for the unknown dataset, the validation is only performed on a limited set of data with fixed amplitudes. It would be helpful to validate the methodology on a larger and more diverse dataset, including EEG signals with varying amplitudes. Statistical analysis results are also suggested to mention.

Furthermore, Figure 3 is missing the x-axis and y-axis labels, which could make it difficult for readers to interpret the data being presented. Adding clear labels would enhance the readability and interpretation of the figure. Finally, the paper would benefit from a more detailed discussion of the reasons behind the selection of the SNR-walls and the SNRs for the different experimental conditions and tasks, which could help to clarify the rationale behind the BCI-Walls methodology and improve the overall quality of the paper.

7. PLOS authors have the option to publish the peer review history of their article (what does this mean?). If published, this will include your full peer review and any attached files.

Reviewer #2: No

---

## [Author Response · Author response to Decision Letter 1]

2 Jul 2023

See the attached file called "loc2.pdf" which contains our response.

---

## [Decision Letter · Decision Letter 2]

9 Aug 2023

BCI-Walls: A robust methodology to predict if conscious EEG changes can be detected in the presence of artefacts

PONE-D-22-30139R2

Dear Dr. Porr,

We’re pleased to inform you that your manuscript has been judged scientifically suitable for publication and will be formally accepted for publication once it meets all outstanding technical requirements.

Kind regards,

Anwar P.P. Abdul Majeed

Academic Editor

PLOS ONE

Additional Editor Comments (optional):

Reviewers' comments:

Reviewer's Responses to Questions

**Comments to the Author**

1. If the authors have adequately addressed your comments raised in a previous round of review and you feel that this manuscript is now acceptable for publication, you may indicate that here to bypass the “Comments to the Author” section, enter your conflict of interest statement in the “Confidential to Editor” section, and submit your "Accept" recommendation.

Reviewer #2: All comments have been addressed

2. Is the manuscript technically sound, and do the data support the conclusions?

Reviewer #2: Yes

3. Has the statistical analysis been performed appropriately and rigorously? 

Reviewer #2: Yes

4. Have the authors made all data underlying the findings in their manuscript fully available?

Reviewer #2: Yes

5. Is the manuscript presented in an intelligible fashion and written in standard English?

Reviewer #2: Yes

6. Review Comments to the Author

Reviewer #2: I have carefully reviewed the revised version of the manuscript, and I am pleased to say that the author has addressed all the concerns raised during the previous review. The paper has significantly improved in terms of clarity, organization, and overall presentation. The author has also incorporated the suggested changes and provided satisfactory explanations where necessary.

7. PLOS authors have the option to publish the peer review history of their article (what does this mean?). If published, this will include your full peer review and any attached files.

Reviewer #2: No

---

## [Editor Report · Acceptance letter]

16 Aug 2023

PONE-D-22-30139R2 

BCI-Walls: A robust methodology to predict if conscious EEG changes can be detected in the presence of artefacts 

Dear Dr. Porr:

I'm pleased to inform you that your manuscript has been deemed suitable for publication in PLOS ONE. Congratulations! Your manuscript is now with our production department. 

Kind regards, 

on behalf of

Dr. Anwar P.P. Abdul Majeed 

Academic Editor

PLOS ONE